# A Predictive Human Health Risk Assessment of Non-Choleraic *Vibrio* spp. during Hurricane-Driven Flooding Events in Coastal South Carolina, USA

**Alexandra M. Frank \*, Mariana G. Cains**  **and Diane S. Henshel**

O'Neill School of Public and Environmental Affairs, Indiana University, Bloomington, IN 47405, USA;
mgcains@iu.edu (M.G.C.); dhenshel@iu.edu (D.S.H.)
**\*** Correspondence: franalma@iu.edu; Tel.: +1-317-965-3133

**Abstract:** Densely populated, low-lying coastal areas are most at-risk for negative impacts from increasing intensity of storm-induced flooding. Due to the effects of global warming and subsequent climate change, coastal temperatures and the magnitude of storm-induced flooding are projected to increase, creating a hospitable environment for the aquatic *Vibrio* spp. bacteria. A relative risk model analysis was used to determine which census block groups in coastal South Carolina have the highest risk of *Vibrio* spp. exposure using storm surge flooding as a proxy. Coastal block groups with dense vulnerable sub-populations exposed to storm surge have the highest relative risk, while inland block groups away from riverine-mediated storm surge have the lowest relative risk. As Vibriosis infections may be extremely severe or even deadly, the best methods of infection control will be regular standardized coastal and estuarine water monitoring for *Vibrio* spp. to enable more informed and timely public health advisories and help prevent future exposure.

**Keywords:** sea level rise (SLR); storm surge; health vulnerability; septicemia; relative risk model (RRM)



## 1. Introduction

Climate change is projected to increase atmospheric and oceanic temperatures globally with increased microvariability, in turn affecting precipitation patterns, increasing the likelihood of extreme weather events and natural disasters, and accelerating sea level rise (SLR) due to melting glaciers and seawater expansion [1,2]. The effects of climate change are already occurring in the United States as manifested by the increased number and severity of hurricanes, wildfires, and extreme precipitation events. As global weather patterns continue to shift, the weather and secondary phenomena are projected to become even more unpredictable and hazardous [3]. The air and water warming also leads to melting of large ice formations (e.g., glaciers) which is leading to global ocean mean water level increases, known as SLR [4]. As the sea level rises, the 40% of United States citizens living in dense urban coastal areas will be affected by flooding and inundation that can negatively affect living conditions and human health. SLR can increase the prevalence of disease pathogens, such as *Vibrio* spp., further inland from the coastline [5,6].

Flooding due to climate change is expected to be particularly severe on coastlines, because of a combination of SLR, storm surges, and precipitation [7–9]. As the sea level rises and hurricane intensity increases, storm surge flooding and increased precipitation will likely intensify along coastlines already vulnerable to severe flooding. The South Carolina coastline is susceptible to severe coastal flooding due to populated low-lying floodplains and urbanized estuaries [10]. In addition, large population centers along the South Carolina (SC) coastline, like Charleston, Myrtle Beach, and Hilton Head, are subject to flooding and its potential economic and health risks.

Historically, tropical storms and hurricanes occur between June and November, but the severity and intensity of these storms has grown, and May hurricanes are becoming more

common along the Atlantic coast [11]. The severity of flooding is proportional to both the hurricane intensity and subsequent storm surges experienced by cities and regions directly impacted by tropical storms [12]. This flooding, coupled with SLR projected by the International Panel on Climate Change (IPCC) [13], will inundate estuaries characteristic of South Carolina with both salt and freshwater, altering the human landscape and positively or negatively affecting the *Vibrio* spp. habitat depending on the salinity of the water [14,15].

Halophilic bacteria (i.e., bacteria that can survive and grow in saline water), such as *Vibrio* spp., grow best in salty waters typical of estuarine habitats in coastal South Carolina (see Table 1). The warm waters of the southern coastal US also encourage *Vibrio* spp. growth, and higher temperatures in southeastern (USA) coastal waters may alter the acceptable salinity for *Vibrio* spp. habitat [16,17]. In the Northeastern coastal estuaries, the optimal salinity for halophilic species of *Vibrio* spp. is between 15 and 25‰, while along the southeast United States coast, *Vibrio* spp. may thrive in salinities as low as 10‰ [18,19]. This difference in optimal salinity may be related to regional water temperature and nutrient differences [20]. As SLR occurs and ocean water advances further inland, the formerly freshwater habitats and estuaries will become more saline, expanding optimal *Vibrio* spp. habitat. Additionally, the predicted climate change related increases in southeastern United States coastal average water temperature will also help *Vibrio* spp. to thrive. Modeling *Vibrio* spp. has proved difficult because of varying global environmental conditions and the issues with accounting for all varying conditions. Based on the equation and model used [14,15,19], different concentrations of *Vibrio* spp. can be calculated for the same estuarine region. Additional discussion of these modeling challenges for the Charleston Harbor region are detailed in Appendix A.

**Table 1.** Salinity range for *Vibrio* spp. and potential habitat water types.

| Water Type | Salinity Range | Citation |
|---|---|---|
| Fresh Water | <0.5‰ (PSU) [1] | [21] |
| Estuary | 0.5–35‰ | [21] |
| Ocean | 32–37‰ (average 35‰) | [21] |
| *Vibrio* spp. optimal salinity | 15–20‰ | [18] |
| *Vibrio* spp. viable salinity | 5–25‰ | [19] |

[1] PSU = Practical Salinity Units = parts per thousand (‰) = g/kg.

Many genera of *Vibrio* spp. are pathogenic to marine life, and some are virulent human pathogens. *Vibrio* spp. infects victims through ingestion and open wounds [22]; thus, recreational swimmers and people working in brackish waters with even low *Vibrio* spp. concentrations are at an elevated risk of infection. *Vibrio vulnificus* and *Vibrio parahaemolyticus* are two species of *Vibrio* that are a top human health concern because they are capable of causing symptoms ranging from mild skin infection to gastroenteritis to septicemia and death. Victims exposed through dermal contact, particularly through open wounds, are more likely to develop skin infections and septicemia. Once a patient develops septicemia, the patient has a 35% chance of death, and mortality can occur in as little as one to two days [23,24].

The US Centers for Disease Control and Prevention (USCDC) has reported at least one case of Vibriosis in coastal South Carolina every year since they first started monitoring for Vibriosis in South Carolina in 1997, except for the year 2000. The number of cases has been steadily rising, particularly in the past decade (see Appendix B, Table A1, [25,26]). Charleston County, with the longest coastline in South Carolina, had the most cases in 2018 with at least 8 confirmed cases, followed by Dorchester and Beaufort counties (Figure 1). *Vibrio* spp. monitoring data indicates that *Vibrio* spp. grows during the summer months along much of the South Carolina coastline [27], which coincides with the region's hurricane season [28].

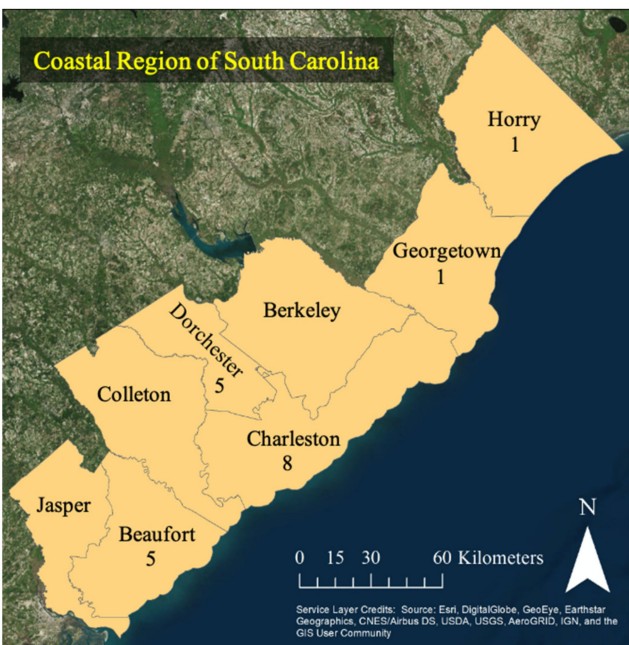

**Figure 1.** Vibriosis cases caused by bacteria *Vibrio* spp. in the coastal counties of South Carolina (USA) for 2018. Absence of number means no reported cases for the county [26].

The main objective of this manuscript is to determine the areas of coastal South Carolina that are inundated during storm surges and could be at higher risk of *Vibrio* spp. infections and mortality due to storm surge impacts. This paper models the predicted overlap between potential exposure to *Vibrio* spp. and denser, vulnerable populations based on age throughout coastal South Carolina utilizing the relative risk assessment model [29]. The initial approach involved modeling *Vibrio* spp. concentrations using Hseih's salinity and temperature model [15]. However, since insufficient salinity data is available for South Carolina, a more predictive approach was used. The relative risk model approach is used in ecological risk assessment when multiple variables are used with units that do not match well to determine areas most vulnerable to environmental threats. In this paper, extent of flooding and socioeconomic vulnerability were used as the main stressor and vulnerability, respectively. In the context of risk assessment, vulnerability is defined as a function of the potential for adverse effects and the ability to cope [30]. Advanced age has been shown to be a strong correlative metric for *Vibrio* spp. mortality, while small children under the age of five have greater relative surface area exposed to flood waters, which increases potential exposure to *Vibrio* spp. [23,24].

## 2. Materials and Methods

### 2.1. Study Area

The geographic focus of this analysis is the eight coastal counties of South Carolina, USA: Horry, Georgetown, Berkeley, Dorchester, Charleston, Colleton, Beaufort, and Jasper, illustrated in Figure 1. These eight counties either border the Atlantic Ocean or are geographically vulnerable to severe flooding from upriver propagation of storm surges and inundation from SLR [13].

Charleston, SC, USA, has an average elevation of 6.10 m (20 ft) above sea level, and lower (~3.05 m (10 ft) elevation) parts of the city bordering the harbor are at risk of flooding with even just 0.30 m (1 ft) of SLR [10]. In addition to flood risk from SLR, the city and watershed also frequently experience heavy rain and flooding from hurricanes and storm surges. For example, in September 2018, Hurricane Florence made landfall in South Carolina and battered the state with heavy rainfall and winds. The highest recorded rainfall was 60.02 cm (23.63 inches) in Loris, SC [31].

### 2.2. Relative Risk Model

Regional risk assessment is an analytical tool used to prioritize management focus on environmental stresses (i.e., stressors) and their potential impact on endpoints of value (e.g., population health, ecological robustness, infrastructure integrity). Landscape scale analysis of a region must take into account the complexity of the components of the landscape, such as the human populations, ecological communities, and both the natural and built infrastructure of the region, thus requiring a method that integrates both across different units of measure and (often) metrics of different scale. Risk assessment methodology requires the identification of an endpoint of value (e.g., vulnerable population) and a route of potential exposure to a stressor (i.e., hazard, e.g., flooding). Next, a conceptual model is developed to causally link the stressor through exposure to the endpoint. Figure 2 illustrates the causal link between storm surge flooding (i.e., brackish water) and potential *Vibrio* spp. exposure for vulnerable populations.

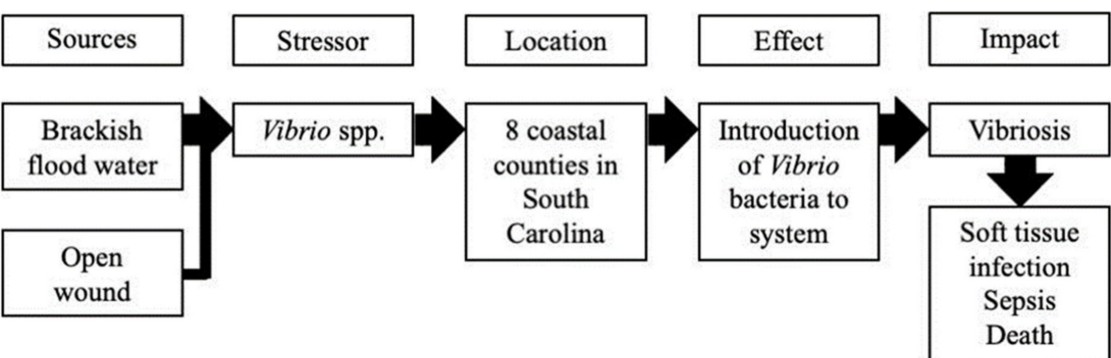

**Figure 2.** Conceptual model paralleling the relative risk model components (source, stressor, location, effect, and impact) and the pathway between the source of *Vibrio* spp. and vulnerable exposure conditions and production of Vibriosis infections.

Over the past 20 years, the relative risk model has been applied in ecological, environmental, and human health risk assessments of physical, chemical, and biological hazards [32–34]. The relative risk model is based on the assumption that regional risk is a function of an endpoint of value being exposed, in space and time, to a stressor/hazard and can incorporate the assessment of multiple levels of impact [35]. For example, a hurricane introduces both wind shear and precipitation as primary stressors, and then causes a storm surge as a secondary stressor and compounding effect. Both the precipitation and storm surge contribute to coastal flooding. The risk is calculated based on potential exposure to flood waters, stressor magnitude (i.e., inundation extent and depth), and vulnerability of the endpoint (i.e., human populations more likely to develop Vibriosis due to flood exposure). When multiple overlapping variables that are characterized by different units are used in a risk assessment, it can be difficult to calculate the overall risk, which is why the relative risk model was selected for this analysis.

The first component of the relative risk model is the Stressor Score, the product of the effect rank and exposure rank, or the depth of flooding and area of inundation, respectively. The Stressor Score is multiplied by the vulnerability rank, which is based on the percentage of at-risk population by age. Ranks are used to transform quantitative data into relative risk rankings based on magnitude of impact. The magnitude of impact is quantified using a scale from 0 to 6. A relative risk rank of zero is 0, low is 2, medium is 4, and high is 6. Any numerical range can be used so long as it is consistent across stressors, exposure, effect, and impact, thus making the calculated risk relative to the assessed components. A detailed explanation of this methodology is provided by Wiegers et al. [34]. The inclusion of GIS (Geographic Information Systems) analysis allowed us to calculate the relative risk for individual areas of land on or near the coastline rather than just the entire coast.

For this paper, age-related vulnerability was used to define vulnerability to flooding, and flooding exposure defined potential exposure to *Vibrio* spp. Age-related vulnerability

and flood exposure were combined to determine risk of dermal exposure to *Vibrio* spp. Areas where health-related flooding vulnerability (health risk due to *Vibrio* spp. exposure) and socioeconomic vulnerability as defined by Cutter et al. (Social Vulnerability Index (SoVI) [36]) and Flanagan et al. (Social Vulnerability Index (SVI) [37]) overlap, will also be areas of higher vulnerability to *Vibrio* spp. infections and adverse health impacts, particularly for subsistence fishers [38]. We assume that hurricane-induced storm surge inundation, which tends to occur in the warmer seasons, produces the appropriate mix of water temperature and salinity to support *Vibrio* spp. growth, given the known presence of *Vibrio* spp. in the South Carolina coastal waterways (see Figure 1 and Table A1).

### 2.3. Data Sources

Spatial data were analyzed, and maps were created using ArcGIS (ESRI, Redlands, CA, USA) [39]. Storm surge modeling data, updated in 2018, were downloaded from the US National Oceanic and Atmospheric Administration [40,41]. Population distribution by block group for 2018 was gathered for the eight coastal counties of South Carolina from the Census Bureau, where a block group is defined as the next level above a census block, but below a census tract in the geographic hierarchy [42,43]. Flooding-related vulnerability is highly age-dependent because young people (who are short) would have difficulty escaping the water and older people (who are more likely to have reduced physical capabilities) would have more difficulty navigating and escaping the water; therefore, vulnerability was assessed based on the percent of the population in these two age groups: ≤5 years and ≥60 years.

### 2.4. ArcGIS and Relative Risk Model Analysis and Visualization

ArcGIS was used to integrate the data and perform a relative risk analysis [29]. Storm surge flooding for each category of hurricane was applied to the eight coastal counties of South Carolina to determine which block groups could potentially be most exposed to *Vibrio* spp. by surface area of inundated land. Relative population vulnerability is determined by the percent of the populace under 5 (short and less mobile in a flood) and over 60 (weaker and also less mobile in a flood). The "tabulate intersection" function was used to calculate the percent of the block group flooded ("Exposure Rank") and the percent of flooded area in each depth category per census block group ("Effect Rank"). Exposure Ranks for a given block group were classified as: 0–33% inundation was ranked as 2, 34–66% exposure was ranked as 4, and 67–100% was ranked as 6. The Effect Rank was weighted based on the percent of flooded area in each depth category per census block group (e.g., (66% × 2) + (23% × 4) + (11% × 6) = 2.66). The Effect Ranks were classified as follows: 0–1 ft of flooding was classified as 2 or low risk, 1–2 ft was classified as 4 or medium risk, and 2+ ft was classified as 6 or high risk. Below 1 foot of water pedestrian movement is impeded, between 1 and 2 feet of inundation movement of both pedestrians and motor vehicles are impeded, and above 2 feet of inundation only boats can move readily through the flood waters [44]. Figure 3 is a flowchart illustrating the geographical analysis and integration of the storm surge, block group, and demographic data.

Using the relative risk model approach, the Exposure Rank and Effect Rank were multiplied to calculate the Stressor Score (= Exposure Rank × Effect Rank) per block group [34]. The final Relative Risk Score was calculated by multiplying the Stressor Score and the ranked percent of vulnerable populations. Vulnerable populations (i.e., ≤5 and ≥60) were ranked as follows: 0% vulnerable population was ranked as 0, $0 \le x < 34\%$ vulnerable population was ranked as 2, $34 \le x < 67\%$ vulnerable population was ranked as 4, and $x \ge 67\%$ was ranked as 6. The final relative risk scores were used to create maps of the geographic distribution of risk to vulnerable populations due to exposure to storm surge flooding as a proxy for *Vibrio* spp. exposure along the South Carolina, USA, coastline (Maps are included in the Results Section).

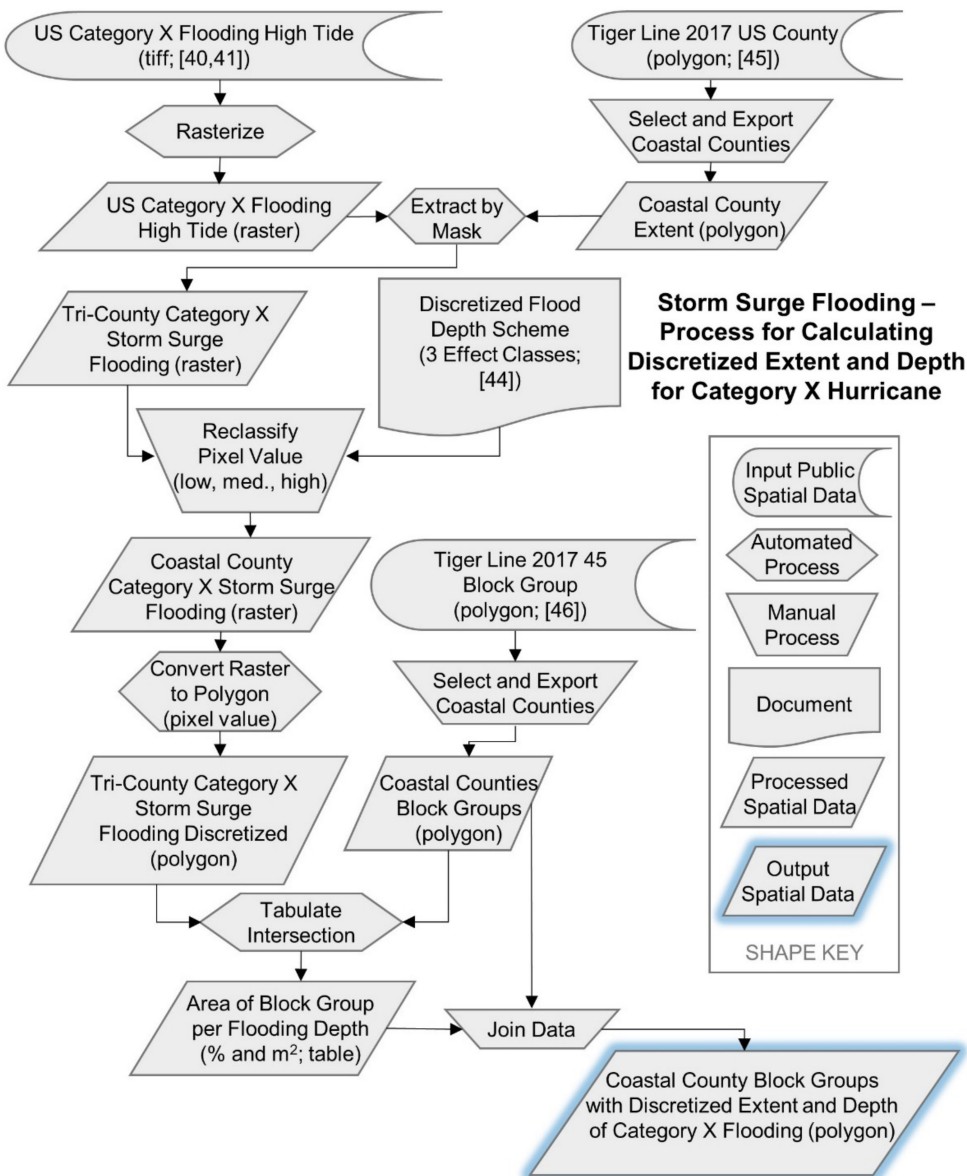

**Figure 3.** Geospatial analysis process for calculating discretized extent and depth of storm surge flooding for any given hurricane category for block groups of coastal counties in South Carolina.

## 3. Results

A typical risk assessment would start with quantifying the amount of the stressor (in this case *Vibrio* spp.) to which the vulnerable population is exposed. As a first approach to quantifying the *Vibrio* spp. concentrations in flood waters, we estimated the probable *Vibrio* spp. concentrations for three estuarine monitoring sites in the Charleston Harbor watershed based on temperature and salinity since the state monitoring data for *Vibrio* spp. in coastal waters is not publicly available. These results are included in Appendix A (see Figure A1). Evaluating the conditions for *Vibrio* spp. growth in the South Carolina coastal waters indicates that summer hydrological conditions are conducive to *Vibrio* spp. growth; however, during Hurricane Florence, *Vibrio* spp. concentrations were likely to decrease for the duration of the hurricane but increase shortly after due to water temperature and salinity fluctuations.

These estimations were not included in the risk assessment because calculations could only be completed for the three monitoring sites in Charleston Harbor and not the full coastline. The USGS only provided salinity data for these three sites in South Carolina

waters. Any other monitoring sites with any indication of ion concentrations only recorded conductivity rather than salinity. Conductivity records any ionic activity and not just salinity. The *Vibrio* spp. model depends on salinity and not on conductivity. The difference between measured salinity and calculated salinity from conductivity measurements hit 5% error below approximate 0.5 PSU salinity (see Figure A2).

Storm surge flooding according to hurricane intensity was mapped across the eight South Carolina counties included in this study as the proxy for *Vibrio* spp. dermal exposure. Figure 4 illustrates the projected storm surge flooding for the eight coastal counties in South Carolina based on the US Sea, Lake, and Overland Surges from Hurricanes model data for Categories 1 through 5 hurricanes [40,41]. The spatial storm surge inundation analysis (Figure 4) indicates that as storm intensity increases, so will storm surge intensity, causing further inland flooding. As more clearly visible in the Category 4 close-up map (Figure 4E), the storm surge flooding gradient rapidly changes from low to high inundation due to the fact that the South Carolina coastline is a relatively flat (low slope) and low-altitude area that remains near sea level, hence the regional moniker "The Lowcountry."

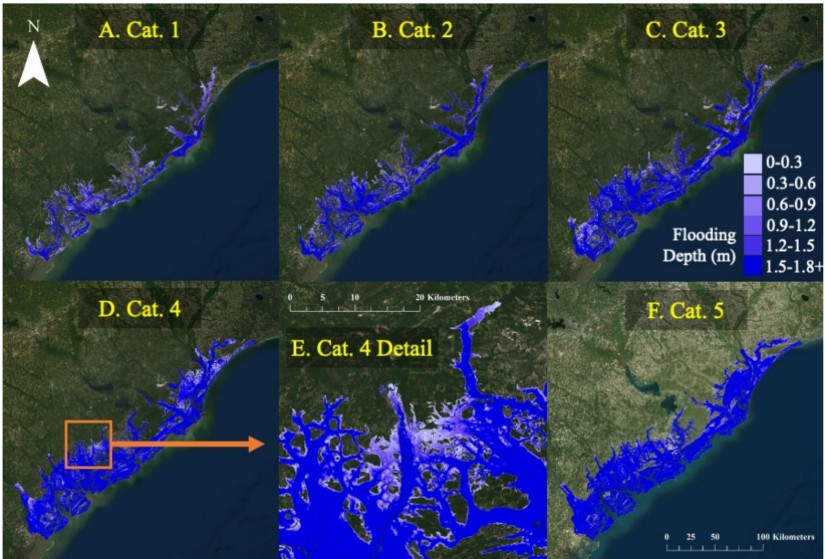

**Figure 4.** Hurricane-induced storm surge inundation maps of the coastal counties of South Carolina based on hurricane intensity: Category 1 (**A**); Category 2 (**B**); Category 3 (**C**); Category 4 (**D**); Category 5 (**F**). Map (**E**) shows a "blow up" of parts of Dorchester and Colleton counties illustrating the storm surge for a Category 4 storm with greater differentiation visible between the flooding depth categories. Service Layer Credits: Source: Esri, Maxar, GeoEye, Earthstar Geographics, CNES/Airbus DS, USDA, USGS, AeroGRID, IGN, and the GIS User Community.

These maps under-represent the maximum modeled flooding depth, which can reach 7.62 m (25 ft) deep around all the rivers and near the coastline for a Category 5 storm. Inundation depths of ≥1.83 m (6 ft) were aggregated into the same dermal exposure category because the vast majority of vulnerable people considered in this study are below 1.83 m (6 ft) in height. The average height for a male in SC, USA, is 1.78 m (5′10″) and female is 1.63 m (5′4″) [45]. Figure 4E visualizes the rapid change in depths from 0.30–1.52 m (1–5 ft) inundation to ≥1.83 m (6 ft) inundation.

Figure 5 shows the total population age distribution by block group used for analysis in the eight coastal counties of South Carolina. Denser populations, shown in dark green, are concentrated around metropolitan areas, Myrtle Beach, Charleston, and Hilton Head. It is particularly noteworthy that some of the denser populations of citizens over the age of 60 are close to the coastline. These citizens are more vulnerable to *Vibrio* spp. exposure during flooding events and are at risk of adverse effects from subsequent infections due to decreased mobility, decreased immune efficiency, and thinner skin [23,46–48].

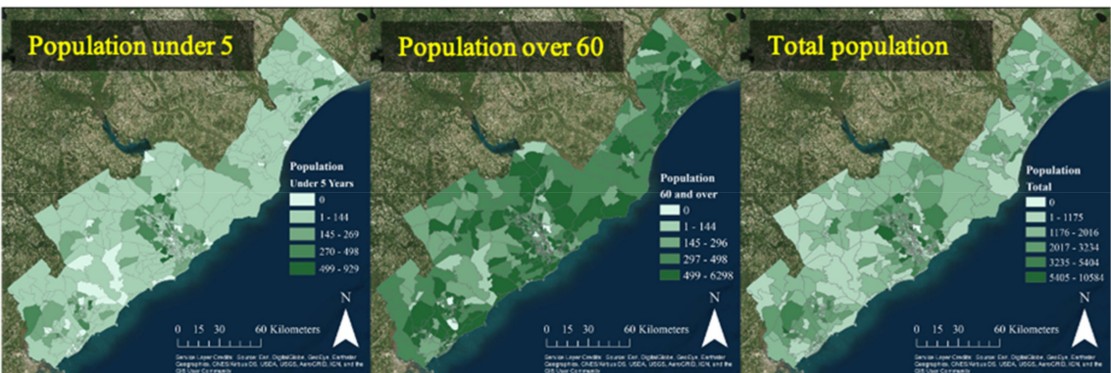

**Figure 5.** Population under 5, over 60, and total population for eight counties of interest in South Carolina, mapped by US Census block group [49,50]. Note that the color scale changes for the total population map versus the two sub–population graphs.

Figure 6A–E shows areas of overlap between highest concentrations of population vulnerable to flooding and susceptibility to *Vibrio* spp. infection (i.e., citizens below the age of 5 and above the age of 60). The darker the blue, the higher the relative risk to *Vibrio* spp. exposure by proxy. High-risk block groups increase in number close to the coastline with hurricane category increases as a result of higher magnitude storm surges associated with the high winds and king tides brought on by the more intense hurricanes. As storm surge inundation increases with hurricane category, the relative risk in inland South Carolina block groups [50] also increases from no risk to low or medium risk until only the furthest inland block groups do not appear vulnerable to storm surge (Figure 6A–E). There are a few outlying block groups in each analysis that have no apparent risk even when surrounding block groups are at low to medium risk. The Census block groups with no vulnerable population have a relative risk of 0 despite the presence of storm surge flooding, as no exposure to vulnerable populations occurs. Without exposure, there is no risk. The Merritt Field (Marine Corps Air Station) in Beaufort County and the 841st Transportation Battalion in Charleston County are locations of zero relative risk due to lack of vulnerable populations, despite being completely surrounded by areas of higher relative risk. These two block groups are white throughout all five categories of hurricane intensity (Figure 6A–E).

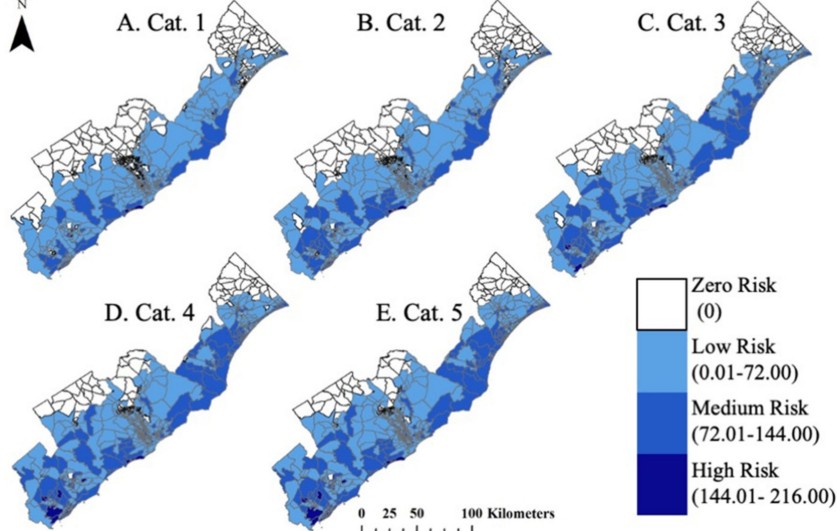

**Figure 6.** (A–E) Images A through E illustrate the spatial Relative Risk Model results of hurricane-induced storm-surge flood risk for the more vulnerable younger (<5 years) and older (>60 years) populations of the eight coastal South Carolina counties, mapped by US Census block group [49,50].

We know that changes in risk in each block group are a direct result of changes in hurricane intensity and resulting storm surge because the vulnerable population by block group remains static for the year analyzed. Other block groups with no risk during low-intensity hurricanes due to minimal encroachment of storm surges have the potential to become low- to high-risk as hurricane intensity increases. Block groups to the north and south of Charleston County have the highest increase in relative risk, as we see some block groups in neighboring counties going from low-risk to high-risk and some going from no risk to medium-risk. The block groups with the greatest changes in risk in Figure 6A–E align with areas of greatest increase in storm surge depth and area in Figure 4A–E.

## 4. Discussion

This relative risk assessment is based on vulnerable population (age-related) and depth and extent of hurricane-induced storm surge flooding and illustrates both the potential risk for Vibriosis for the population of the coastal South Carolina region and the need for further studies on the risk from *Vibrio* spp. exposure posed to coastal communities. The relative risk in this region will increase with the predicted intensification of Atlantic hurricanes and coastal flooding due to climate change. In addition, the coastal South Carolina population is both increasing and aging [51,52]. Inland block groups within coastal South Carolina counties currently at zero to no risk may remain at low risk of flooding and therefore low risk of *Vibrio* spp. exposure due to a lack of inland propagation storm surges. However, the modeled storm surge flooding data used in the relative risk analysis did not include sea level rise (SLR) in its calculation, meaning the full extent of climate change effects is not captured in this analysis. Over the period of 1920 to the present, global SLR has increased approximately 18 cm (7 inches [1]), while "nuisance flooding" in Charleston has increased from rare (none to a few times a year in 1920) by at least an order of magnitude [53,54]. In 2019, for example, Charleston experienced 77 days of nuisance flood events [55].

The distinction between no risk due to no vulnerable population and no risk due to no storm surge flooding is important because as sea levels rise due to climate change, storm surges will intensify and propagate further inland, exposing more block groups to *Vibrio* spp. This risk will likely increase from no risk to low/medium risk in the block groups with vulnerable populations, and block groups already at risk will likely experience increased risk. This exacerbates the risk of *Vibrio* spp. exposure on inland vulnerable populations that may need to wade through storm surges to escape dangerous flooding at risk where they previously were not at risk. Other social determinants of health like occupation and income can also affect vulnerability to *Vibrio* spp. and could be used in future analysis [37].

The South Carolina coastline on the Atlantic Ocean is a popular vacation destination for many Americans, so there are many state parks along the coastline, including the Edisto Beach State Park and the Huntington Beach State Park [56]. These state parks may contribute to the lack of population in multiple coastal block groups. In addition to state parks, the Waccamaw National Wildlife Refuge, Francis Marion National Forest, and 841st Trans Battalion in Charleston are all areas with little to no population that would affect the relative risk calculations. The block groups with no vulnerable populations due to the presence of state parks, wildlife refuges, and military facilities would be more likely to have increased vulnerable populations if the land uses change.

Flooding is used as a proxy for *Vibrio* spp. exposure and infection in this analysis, so civilians most vulnerable to both flooding and infection are considered at risk. Children under five are very vulnerable to flooding because even at a low-depth floodwaters can be incredibly hazardous to them. A greater surface area of a child's skin will be exposed to bacterial infection in a lower depth than will be exposed for an adult. Adults over 60 are at greater risk of more severe Vibriosis infections leading to complications such as septicemia and death. Older adults also have thinner, more fragile skin and are more vulnerable to having pre-existing or concurrent wounds when wading through floodwater. In addition, adults over the age of 60 have a higher population prevalence of comorbidities like liver

and heart disease. While underlying health conditions can enable a worse outcome for *Vibrio* spp. infections, health of an individual cannot totally predict clinical outcome, so not all vulnerable populations were included in this study [46,57].

The South Carolina USA population over the age of 60 is expected to more than double by the year 2030, as projected by the South Carolina Revenue and Fiscal Affairs Office. In addition, the population has more than doubled since 1970 (2,590,516 to 5,148,714), with a majority of the change coming from the 60 and older age group [52]. An aging population experiencing increased relative risk of exposure to *Vibrio* spp. by flooding proxy will both place greater strain on the healthcare system and negatively impact the population itself.

*Vibrio* spp. concentrations are projected to increase to as much as four times the levels they are now in already hospitable areas such as South Carolina [14]. An interesting trend observed in this research was that while many different environments lining coastlines are hospitable to *Vibrio* spp., some models developed in specific regional climates cannot be extrapolated to other climates. For example, the model developed by Lanerolle in the Chesapeake Bay area was used initially in this analysis, and every output predicted a greater than 99.9% probable presence of *Vibrio* spp. using Charleston Harbor water quality data as the input [58]. While some models can be extrapolated, like the model used by the European Centre for Disease Prevention and Control (ECDC) for forecasting global suitability for *Vibrio* spp., many models are geographically specified and cannot be used for extrapolation, thus increasing the need for further studies on *Vibrio* spp. exposure and infection.

As climate change and rising sea levels expose more vulnerable populations to *Vibrio* spp., the conditions for *Vibrio* spp. growth and survival also become more favorable. Impending SLR is an important factor to consider in *Vibrio* spp. exposure and risk models. SLR was not quantified in this study because the US National Hurricane Center's National Storm Surge Hazards Maps did not include projected sea level rise. The modeling capabilities needed for such an integrated analysis were beyond the scope of this paper. In recently published First Street Foundation data, storm surge, precipitation, and SLR are incorporated into the same model and produced a dataset that encompasses more the flood factors associated with climate change [59]. Future collaborations with First Street Foundation's Flood Lab could provide the additional information needed to model the *Vibrio* spp. concentrations inclusive of climate change compounded flooding.

The influence of SLR on local *Vibrio* spp. concentrations may be variable due to local geophysical differences and variability in weather patterns. In South Carolina, higher ocean temperatures, rising sea levels, and more severe hurricanes will drive the warm brackish waters that sustain *Vibrio* spp. populations further inland. However, this phenomenon depends on salinity. Heavy inland precipitation during hurricanes can result in freshwater being driven out to the sea through the river system, thus temporarily reducing both salinity and viable *Vibrio* spp. concentrations in the estuaries and along the coasts [14,60–62]. The current analysis utilized a relative risk approach in large part because salinity data for the South Carolina coastal watershed is extremely limited. This weather driven change in salinity affecting *Vibrio* spp. concentrations could be modeled if salinity was monitored consistently along the coast.

While the lack of dose-response curves for dermal exposure to *Vibrio* spp. limits the specificity of this risk assessment, a relative risk assessment identifying areas of high vulnerability and potential risk can be achieved. Epidemiological monitoring is currently problematic given that by the time people present with Vibriosis, the sources and concentrations of exposure can no longer be definitely identified. In a typical human health risk assessment, a dose-response curve is used to quantify the probable health effects posed to humans from exposure to the stressor of interest. The lack of *Vibrio* spp. dose-response curves limits the effectiveness of this research in translating the concentration of *Vibrio* spp. in floodwaters to human health risk for vulnerable populations. Possible approaches to developing dose-response data include human studies, animal studies, or meta-analysis.

However, due to ethical considerations about the severity of *Vibrio* spp. infections, both human and animal studies are not recommended. Meta-analysis of global cases of *Vibrio* spp. exposure and infection is the preferred method of developing exposure-response curves. For any future meta-analysis, consistent *Vibrio* spp. monitoring alongside existing *Escherichia coli* and *Vibrio cholerae* monitoring will be essential. Public health officials in susceptible regions (e.g., warm coastal waters) should develop a mandatory, ongoing, standardized *Vibrio* spp. monitoring system due to the potentially severe health impacts of *Vibrio* spp. growth. Monitoring should be paired with a global warning system as is currently used for *E. coli* contamination, harmful algal blooms, and severe air pollution.

## 5. Conclusions

Climate change is adversely affecting human health, in part by increasing the viable habitat for disease pathogen growth, contributing to greater frequency of disease transmission [63]. The results of this study indicate that coastal census block groups with dense vulnerable sub-populations exposed to storm surge have the highest relative risk, while most inland block groups away from riverine-mediated storm surge have the lowest relative risk. The higher risk for densely populated coastal areas is driven by vulnerability, while the risk for less populated areas is lower due to a lack of exposure. The pathogenic *Vibrio* species are a noted public health concern in the recent Lancet Countdown Report on Health and Climate Change [63]. As climate and weather conditions in South Carolina create more hospitable estuarine conditions for *Vibrio* spp. and increase the range and concentrations of the bacteria, more people (and seafood) will become infected.

Public health and environmental managers will need to develop new management strategies to prevent these negative public health externalities of climate change. As Vibriosis infections may be extremely severe or even deadly (35% mortality rate after developing septicemia [23,24]), the best methods of infection control will be regular standardized coastal and estuarine water monitoring for *Vibrio* spp. to enable more informed and timely public health advisories and help prevent future exposure.

Prevention requires providing the public with information before or as exposure potential increases and needs to include actionable advice to be most effective [64]. The European Centre for Disease Prevention and Control (ECDC) *Vibrio* Map Viewer Geoportal provides near real-time (up through the day before) and predictive (five-day forecast) estimates of *Vibrio* spp. risk based on remotely sensed sea surface temperature and sea surface salinity using a model standardized to the Baltic Sea [65–68]. As indicated in the description about the *Vibrio* Map Viewer, this system can only be used for prediction of *Vibrio* spp. in regions for which the *Vibrio* spp. growth model has been standardized.

The US Centers for Disease Control and Prevention (US CDC) tracks waterborne *Vibrio* spp. infections using the Cholera and Other *Vibrio* Illness Surveillance (COVIS) system [69], the National Notifiable Diseases Surveillance System (NNDSS) [70], and the Top of Form.

National Outbreak Reporting System (NORS) [71]. None of the US surveillance systems for *Vibrio* spp. provide up to date information and cannot be used as the basis for a warning or prevention system, like the ECDC *Vibrio* Map Viewer. We recommend that the US CDC fund studies to standardize *Vibrio* spp. growth models based on remotely sensed sea surface temperature and salinity data normalized to salinity monitoring data, and either feed that model to the ECDC *Vibrio* Map Viewer or create a similar *Vibrio* Map Viewer calibrated to U.S. water conditions.

**Author Contributions:** A.M.F. was the primary author, primary researcher, and primary producer of figures and graphs. A.M.F. and M.G.C. conceptualized the project. M.G.C. was the direct supervisor for data acquisition and analysis, ArcGIS analysis and map production, produced some figures, and was a co-editor. D.S.H. was the direct supervisor for writing, was the research consultant, produced one graph, and was a co-editor. All authors have read and agreed to the published version of the manuscript.

**Funding:** This research received no external funding.

**Institutional Review Board Statement:** Not applicable.

**Informed Consent Statement:** Not applicable.

**Data Availability Statement:** The input data for this study are all publicly available through the links provided in the references.

**Acknowledgments:** The authors thank Madison Howell and Kate Zhang for advice, suggestions, and support. The authors also appreciate the reviewers' comments; addressing the comments strengthened the paper. Maps throughout this article were created using ArcGIS® software by Esri. ArcGIS® and ArcMap™ are the intellectual property of Esri and are used herein under license. Copyright © Esri. All rights reserved. For more information about Esri® software, please visit www.esri.com.

**Conflicts of Interest:** The authors declare no conflict of interest.

## Appendix A

*Vibrio* spp. concentration calculations for three USGS water monitoring sites in Charleston Harbor watershed (Charleston, South Carolina, USA).

### Appendix A.1 Methods

Data were collected from USGS hydrologic unit codes (HUC) 021720709 (1) and 021720710 (2) in watershed 03050201 (HUC name "Cooper"), and 02172053 (3) in watershed 03050202 (HUC name "South Carolina Coastal"). Climate-related water data (2018; 15-min intervals) used to model the *Vibrio* spp. concentrations were downloaded from the United States Geological Survey (USGS) website on water quality [17].

*Vibrio* spp. concentrations were estimated using the concentration prediction equations based on water temperature and salinity derived by Hsieh et al. [15] applying water temperature and salinity data from 2018 for Charleston, SC [17]. The Hseih model coastal data was built on North Carolina coastal data, and no model for all *Vibrio* spp. is available for South Carolina. The following equation was used to predict *Vibrio* spp. concentrations:

$$-\log_{(Vibrio\,spp.)} \text{Colony Forming Units [CFUs]} = -0.304 + (0.116 \times S) + (0.0739 * T) \qquad \text{(A1)}$$

where S = Salinity (ppm) and T = Temperature (°C)

USGS site data from within the Charleston Harbor Watershed were selected because the datasets included temperature and salinity profiles for the full year. The year chosen (2018) had two intense weather events that led to severe flooding (Hurricane Florence and Hurricane Michael), and two less severe flooding events, according to the US National Weather Service Significant Weather Archive [72]. For this analysis, only the month in which Hurricane Florence landed (September) was analyzed in detail. The temperature and salinity data were applied to the Hsieh model to estimate the predicted *Vibrio* spp. CFUs for the full year and for the two weeks before and after Hurricane Florence (see Figure A1). As the USGS reported data for every 15 min during and after the storm event, our model predicted Vibrio concentrations every 15 min, which were plotted as time vs. Colony Forming Units (CFUs).

### A.2. Results

Figure A1 shows example *Vibrio* spp. concentrations calculated every 15 min for the full year for site 1 (021720709). The insets in the upper right corner of the graph detail the period in 2018 during which Hurricane Florence made landfall in South Carolina (28 September 2018 to 18 October 2018). Hurricane Florence dynamically altered both the temperature and the salinity at each site, altering the predicted *Vibrio* spp. concentrations for up to a week before landfall and up to two weeks after moving on for the more central harbor site. Predicted *Vibrio* spp. concentrations changed within three days of Hurricane Florence landfall for the edge of the harbor and the up-river sites (data not shown) which are not as directly influenced by the ocean currents as is the central harbor. After Hurricane Florence made landfall, the central harbor site returned to pre-hurricane

*Vibrio* spp. concentration conditions faster than the edge of the harbor and the up-river sites, up to three weeks after Hurricane Florence moved inland.

Site 1 (021720709; central harbor) shows high predicted *Vibrio* spp. concentrations in the summer, with peak concentration on 14 July at 497,348 CFUs/100 mL (Figure A1). The effects of Hurricane Florence appear to be a sharp decrease in predicted *Vibrio* spp. concentrations seen at the times leading up to and where Hurricane Florence made landfall. The predicted concentrations of *Vibrio* spp. then increased afterwards.

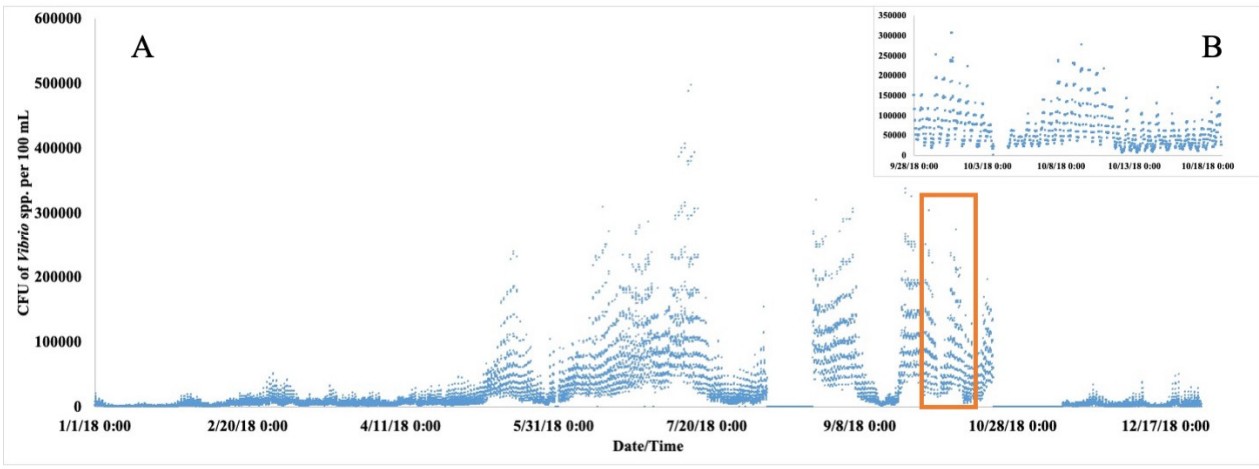

**Figure A1.** Calculated *Vibrio* spp. concentrations for site 021720709 CFUs/100 mL for 2018 including Hurricane Florence (top right corner inset, (**B**)). (**A**) Calculated *Vibrio* spp. concentrations (in CFUs) for site 021720709 for all of 2018. Each point represents a 15-min detection window. (**B**) Calculated *Vibrio* spp. concentrations (in CFUs) for site 021720709 for the period during and after Hurricane Florence. The time window for (**B**) is indicated in (**A**) by the red box.

The USGS water monitoring program does not consistently measure all of the same metrics at all of the sites. Salinity data is particularly sparse for the USGS South Carolina water quality data, having been measured only for three sites in the Charleston Harbor watershed. Conductivity, however, is measured at most South Carolina water monitoring sites.

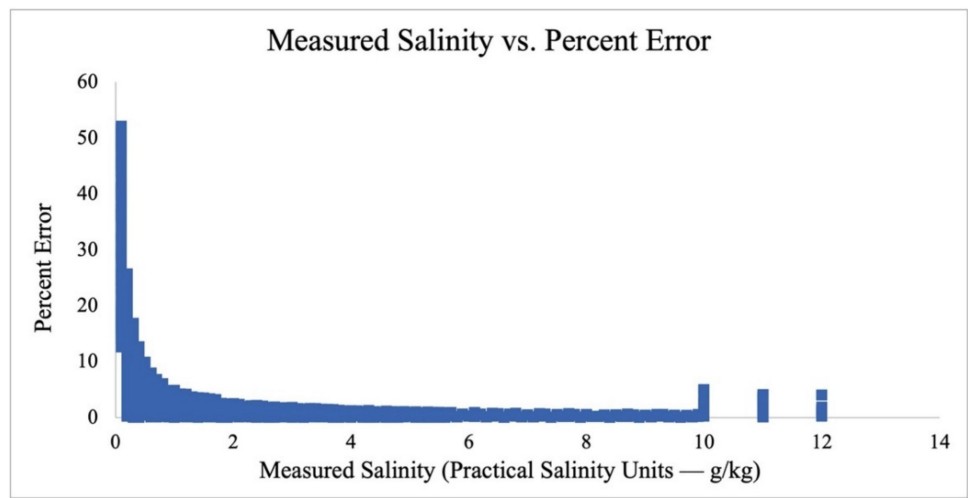

**Figure A2.** Plot of measured salinity against the percent error of salinity calculated from conductivity compared to the salinity directly measured at the same site (site 021720709) at the same time. Percent error is the difference between measured salinity and calculated salinity divided by the measured salinity. Data for conductivity and salinity were measured every hour throughout 2018. The full 2018 hourly dataset was used for this analysis.

Optimal growth conditions for *Vibrio* spp. in water seem to be based specifically on salinity, and not on all of the ions that might be assessed in conductivity measurements [15,16]. In order to calculate *Vibrio* spp. concentrations from salinity and temperature, conductivity needs to be converted to salinity because most USGS water monitoring sites in South Carolina record conductivity but not salinity. We calculated salinity from conductivity for a site for which we had co-collected salinity and conductivity data using the equation used for Practical Salinity Scale of 1978 and then compared the calculated against measured values (Figure A2). Details about the development and parameterization of the equation are documented in Lewis [73]. The generic equation to calculate salinity from conductivity is as follows:

$$S(‰) = a_0 + a_1 K_{15}^{1/2} + a_2 K_{15} + a3 K_{15}^{3/2} + a_4 K_{15}^2 + a_5 K_{15}^{5/2} \tag{A2}$$

To assess the appropriateness of using the calculated salinity (Equation (A2)) for a classical dose-response-based risk assessment, we determined the accuracy of the calculated salinity by comparing the calculated salinity against the measured salinity for a single monitoring site in Charleston Harbor, SC, USA, using all of the data collected hourly over a single year ($n$ = 4990 data points). The percent error for calculated salinity exceeds 5% at salinities below approximately 0.5 PSU. As the salinity decreases below this inflection point (~0.5 PSU), the percent error exponentially increases. As the estuarine salinity concentrations were less than 0.5 PSU for approximately a third of the measurements, we determined that using salinity calculated from conductivity would not be accurate enough to use in classic risk assessment models.

## Appendix B

Recorded Vibriosis cases in South Carolina, USA for 2018.

**Table A1.** The following data on confirmed Vibriosis cases were collected by the CDC's COVIS monitoring program and obtained via personal communication from Claire Youngblood and Marya Barker [26]. The cases are for the year 2018: the eight coastal or near-coastal counties included in this analysis typically comprise over half of the total cases of Vibriosis in South Carolina.

| County | Cases 2014 | Cases 2015 | Cases 2016 | Cases 2017 | Cases 2018 |
|---|---|---|---|---|---|
| Beaufort | 3 | 2 | 5 | 4 | 5 |
| Berkeley | 0 | 0 | 2 | 1 | 0 |
| Charleston | 6 | 1 | 6 | 9 | 8 |
| Colleton | 0 | 0 | 1 | 1 | 0 |
| Dorchester | 0 | 0 | 1 | 3 | 5 |
| Georgetown | 0 | 0 | 0 | 1 | 1 |
| Horry | 3 | 3 | 1 | 2 | 1 |
| Jasper | 1 | 1 | 0 | 0 | 0 |
| 8 County Totals | 13 | 7 | 16 | 21 | 20 |
| South Carolina Totals | 18 | 11 | 22 | 37 | 39 |
| Percent of SC Cases | 72% | 64% | 73% | 57% | 51% |

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
