# Peer review of "A Predictive Human Health Risk Assessment of Non-Choleraic Vibrio spp. during Hurricane-Driven Flooding Events in Coastal South Carolina, USA"

_atmosphere, doi:10.3390/atmos12020269_

Round 1

Reviewer 1 Report

This is an overall welll-written manuscript, where the authors have used a previously developed relative risk assessment model in order to determine vulnerability to Vibrio spp exposure after hurricane and storm related flooding in South Carolina.

Methodology is well described and scientifically sound and results are explicitly presented. My only major comment is that some parts of the manuscript are often too long, information is sometimes redundant and this compromises the density and flow of the text. For example, the description of methods used at the last paragraph of the introduction could be incorporated in the methods section. 

One more comment is the limitation that salinity data was not available for all studied areas, since some monitoring sites only reported conductivity data, thus potentially compromising the strength of calculated predictions.

Also, why did the authors not incorporate seal level rise into their storm surge flooding model, since Vibrio spp' presence also depends on the rising sea level, (due to change in temperature and estuarine conditions affecting the abundance of plankton and the presence of Vibrio spp).

Were there any previous quantitative data on Vibrio spp abundance associated with flooding due to storm surges in the areas studied? Do we know what the change in Vibrio spp populations has been in these areas so far?

Do the authors believe that social determinants of health like socioeconomic status, or type of work activity (i.e. fishing, outdoor professions) could also be important for assessing population vulnerability to flooding related hazards?

Finally, the authors can comment in a little more detail on the preventive measures that public health officials could take in order to alert persons living in coastal areas when risk becomes moderate or high. For example active surveillance for Vibrio spp and quantititavive assessment in estuaries, alerts when conditions are favorable for the emergence of Vibrio spp (temperature, sea level rise, salinity), targeted messages to people with comorbidities who may have worse outcome after vibrioses (i.e. patients with liver disease, older age).

Reviewer 2 Report

I think that overall the manuscript is written in a decent and straightforward manner. 

Please find my comments below on each section of the manuscript.

Abstract:
Captures the summary of the findings.

Introduction:
Line 25: may read better if broken down into two or more sentences.

Materials and Methods:
Well explained, and straightforward.

Results:
Presented in a relatively easy-to-understand manner.

Discussion:
Appropriate.

Conclusions:
Appropriate.

Overall, I think that this manuscript can be accepted in its present form. I just have one very minor suggestion in the introduction as mentioned above. The authors did a decent job of explaining different components of the model and the terms used within through flowcharts. The methods are straightforward and should be relatively easy to understand. Although vibriosis can be caused due to ingestion of raw/undercooked seafood as well, this study models the relative risk of Vibrio spp. to people of specific age groups in South Carolina under flooding conditions. Although the findings of this study are not unexpected, they fill a gap in our understanding of the ecology/epidemiology of vibriosis and predict the risky areas for vibriosis in South Carolina. The authors also pointed out the limitations of the study later in the discussion section, which is much needed.
